# Remdesivir Alone or in Combination with Monoclonal Antibodies as an Early Treatment to Prevent Severe COVID-19 in Patients with Mild/Moderate Disease at High Risk of Progression: A Single Centre, Real-Life Study

**DOI:** 10.3390/vaccines11020200

**Published:** 2023-01-17

**Authors:** Riccardo Scotto, Antonio Riccardo Buonomo, Antonio Iuliano, Maria Foggia, Alessia Sardanelli, Riccardo Villari, Biagio Pinchera, Ivan Gentile

**Affiliations:** Department of Clinical Medicine and Surgery—Section of Infectious Diseases, University of Naples Federico II, Via S. Pansini 5, 80131 Napoli, Italy

**Keywords:** COVID-19, SARS-CoV-2, remdesivir, monoclonal antibodies, hospitalization, fragile

## Abstract

Early treatment with antivirals against SARS-CoV-2 infection can prevent the onset of severe COVID-19 in fragile and immunocompromised patients. In this real-life, prospective, observational study, we evaluated efficacy and safety of a 3-day early treatment with remdesivir in adult and fragile patients with a diagnosis of SARS-CoV-2 infection who referred to the COVID-19 early treatment service of Infectious Diseases Unit of University of Naples Federico from 10 January 2022 to 31 March 2022. The included patients could be treated with either remdesivir alone or with remdesivir plus a monoclonal antibody with activity against SARS-CoV-2. Among the 62 included patients, we showed low rates of hospitalization (8%), increase in oxygen supplementation (3.2%), ICU admission (1.6%) and death (1.6%). The rate of disease progression was 8% and it was similar in patients treated with remdesivir alone or in combination with monoclonal antibodies (6.7% and 9.4%, respectively; *p* = 0.531). The rate of adverse drug reaction was low and similar in the two groups (13.3% in patients treated with remdesivir, 15.6% in patients treated with the combination; *p* = 0.543). Most common adverse events were headache and fever. In conclusion, in our cohort of patients at a high risk of worse COVID-19 outcomes, an early course of remdesivir showed low rates of disease progression and adverse drug reactions.

## 1. Introduction

Natural history of COVID-19 dramatically changed after the introduction of global vaccination. However, COVID-19 still represents a public health concern due to the selection of new variants able to escape vaccine-induced immunity, the high rate of breakthrough infections (i.e., infections occurring in vaccinated patients), the presence of frail patients at risk for severe evolution of the disease despite vaccination, and the non-negligible percentage of patients defined humoral non-responders to vaccination. With the treatment of influenza (another acute respiratory infection) taken as a paradigm [1,2], administration of antivirals against SARS-CoV-2 was tested in the first days after symptoms onset. This approach for COVID-19 included different therapeutic strategies (monoclonal antibodies, oral antivirals, such as molnupiravir or nirmatrelvir/ritonavir, or the intravenous antiviral remdesivir) [3,4]. Remdesivir is a direct-acting nucleotide prodrug inhibitor of the SARS-CoV-2 RNA-dependent RNA polymerase. It showed potent nanomolar activity in primary human airway epithelial cells [5]. Currently, apart from pivotal trials, few data on the efficacy of early treatment with remdesivir in outpatients with SARS-CoV-2 are available. In the PINETREE trial, the risk of COVID-19-related hospitalization or death from any cause by day 28 was 87% lower in the remdesivir group than in the placebo group during the first seven days of symptoms and with at least one pre-existing risk factor for progression to severe COVID-19. Risk factors considered in the PINETREE trial were hypertension, cardiovascular or cerebrovascular disease, diabetes mellitus, obesity (BMI ≥ 30), immunodeficiency, mild or moderate chronic kidney disease, chronic liver disease, chronic lung disease, current cancer, or sickle cell disease. However, among enrolled patients, the vast majority reported diabetes mellitus, obesity, and hypertension as risk factors (62%, 55%, 48%, respectively) [6]. The major limitations of this trial, which preclude its generalizability in everyday clinical practice, are the inclusion of unvaccinated patients and the lack of patients with non-omicron variants. Moreover, current authoritative guidelines on the management of patients with COVID-19 recommend the use of early three-day treatment course with remdesivir in patients at high risk for severe disease progression (e.g., the Infectious Diseases Society of America [7] guidelines and the Italian Society of Infectious and Tropical Diseases guidelines [8]). However, doubts on the efficacy of remdesivir against SARS-CoV-2 related to the reports of resistance in case reports and in vitro studies have been raised [9,10,11]. Therefore, after the FDA (Food and Drug Administration) emergency approval of Remdesivir for this indication, there is an urgent need of real-life data with the inclusion of vaccinated patients and infected with omicron variants.

## 2. Material and Methods

We conducted a prospective real-life study in all adult patients with a diagnosis of SARS-CoV-2 infection who referred to the COVID-19 early treatment service of Infectious Diseases Unit of University of Naples Federico II and treated with a 3-day course of remdesivir from 10 January 2022 to 31 March 2022. We did not set exclusion criteria, but patients had to comply to the eligibility criteria established by the Italian Drug Agency (AIFA, Agenzia Italiana del Farmaco) to be applied for early 3-day treatment with remdesivir. In detail, AIFA eligibility criteria were the following [12]:-Diagnosis of COVID-19.-No hospitalization due to COVID-19.-No oxygen supplementation therapy due to COVID-19.-COVID-19 related symptoms with onset within 7 days.-Presence of risk factors for progression to severe COVID-19, namely active onco-haematologic disease, chronic kidney disease (excluding patients with estimated glomerular filtration rate [eGFR] < 30 mL/min/1.73 m^2^), severe bronchopulmonary disease, inherited or acquired immunodeficiency, obesity (body mass index ≥ 30), severe cardiovascular disease, chronic hepatic disease.-Absence of severe liver injury (denoted by liver transaminases three times higher the upper normal value) and estimated glomerular filtration rate < 30 mL/min/1.73 m^2^.

Both outpatients and inpatients were included, with the latter being necessarily hospitalized for reasons different from SARS-CoV-2 infection. Patients must have had a confirmed SARS-CoV-2 infection, namely a positive molecular SARS-CoV-2 oro-pharyngeal swab; they also had to provide consent to treatment, signing a specific informed consent form. All the included patients received remdesivir 200 mg intravenously, diluted in 250 mL of isotonic saline solution at day 1, and remdesivir 100 mg intravenously diluted in 250 mL of isotonic saline solution at days 2 and 3. Patients could also receive an additional treatment with monoclonal antibodies (mAbs) against SARS-CoV-2, at medical site staff discretion. Available mAbs which could be administered in association with remdesivir were casirivimab/imdevimab and sotrovimab. Casirivimab/imdevimab was administered as a single administration at a dosage of 600 + 600 mg intravenously, while sotrovimab was administered as a single administration at a dosage of 500 mg intravenously. Before treatment administration, all patients were asked to perform a blood sample collection for SARS-CoV-2 IgG dosing. Refusal to perform the blood sample collection was not considered an exclusion criterion, to reflect the real-life nature of the study.

After treatment completion, outpatients were invited to contact the medical staff to refer any change in their clinical condition, including possible adverse drug reactions (ADRs). Patients who referred a worsening in their clinical conditions were invited in our ward to perform a follow-up visit and were hospitalized if necessary. In case of hospitalization before completion of remdesivir treatment, this was continued. ADRs were recorded daily by medical staff among inpatients, while outpatients (as well as inpatients after discharge), where contacted by telephone every 72 h to collect any ADR up to 21 days from treatment completion. Only ADRs related to remdesivir or mAbs administration, as judged by the medical staff, were recorded. ADRs possibly related either to drug administration or to SARS-CoV-2 infection were also recorded. The prevalence of occurrence of the following unfavourable outcomes was collected: hospitalization (among outpatients), need or increase in oxygen supplementation, admission in intensive-care unit (ICU), and death. Need or increase in oxygen supplementation was defined as the need for oxygen therapy in patients who did not perform oxygen therapy at T0, or the increase in oxygen supplementation in patients already in oxygen therapy at T0. A combined outcome named “Progression” of COVID-19 was defined as the presence of at least one unfavourable outcome (namely hospitalization, increase in oxygen supplementation, ICU admission, and death). This study was conducted according to the world medical association declaration of Helsinki on ethical principles for medical research involving human subjects. The study protocol was approved by the local ethical committee (Prot. N. 98/2022 ID: N. 1032)

### Statistical Analysis

All the variables were tested for parametric/non-parametric distribution with the Kolmogorov–Smirnov test. Comparisons between categorical dichotomous variables were performed with the χ2 test (or with Fischer’s exact test when applicable), while comparisons between mean of quantitative variables were conducted with Student’s t-test (parametric variables) or the Mann–Whitney U test (non-parametric variables). For all the tests, a *p*-value of <0.05 was considered for significance. IBM SPSS© version 27 was used for statistical analysis.

## 3. Results

In total, 62 patients were included in the study; 30 (48.4%) were treated with remdesivir alone (Group 1), while 32 (51.6%) were treated with the combination remdesivir + mAbs (Group 2). Namely, 13 (40.6%) and 19 (59.4%) patients in the combination treatment group received casirivimab/imdevimab and sotrovimab, respectively. Considering the whole sample, about half of the patients were male (33, 53.2%) and median age was 66 years (IQR: 49–76). Most patients (55, 88.7%) had at least one comorbidity, with immunodeficiency being the most frequent one (36 patients, 58.1%) (Table 1). About three quarters (75.8%) of the included patients received SARS-CoV-2 vaccination, while 34 (54.8%) out of 49 patients who performed the blood sample collection had positive SARS-CoV-2 IgG. Patients in Group1 more frequently had diabetes (16.7% vs. 0.0%, *p* < 0.05) and cardiovascular disease (40.0% vs. 6.3%, *p* < 0.01) compared with patients in Group 2, while the latter more frequently showed a negative SARS-CoV-2 serology at admission (37.5% vs. 10.0%, *p* < 0.05). No other differences were recorded between the two groups (Table 1). No differences in clinical outcomes were observed between the two treatment groups. Overall, the rates of hospitalization, oxygen support escalation, ICU and death were very low. In fact, the hospitalization rate (for outpatients) and the overall increase in oxygen supplementation were 8% and 3.2%, respectively. Finally, no differences in the time from positive swab to negative swab, and from treatment initiation and negative swab, between the two treatment groups were recorded.

Overall, five patients (8%) showed progression of COVID-19 (defined as at least one among hospitalization, increase in oxygen supplementation, ICU admission, and death). No factors associated with progression were found (Table 2). The rate of progression was similar among patients treated with remdesivir alone (n = 2, 6.7%) and those treated with the combination of remdesivir + mAbs (n = 3, 9.4%, *p* = 0.531).

Among patients who performed serum SARS-CoV-2 IgG dosing (n = 49), no differences in the rate of disease progression were found between those with negative serology and patients with positive serology (33.3% vs. 66.7%, *p* = 0.349). Three out of five patients who showed progression were tested for SARS-CoV-2 IgG, one of them was treated with remdesivir alone (5%), while two were treated with the combination remdesivir + mAb (6.9%, *p* = 0.639). Globally, only nine ADRs were recorded among nine patients, four (13.3%) in remdesivir-only treated patients, and five (15.6%) in patients treated with combination, respectively (*p* = 0.543) (Table 3). The most common ADRs were headache and fever, which both occurred in four patients (6.5%). Headache only occurred in patients who received mAbs (*p* = 0.064).

## 4. Discussion

This is one of the first studies reporting real-life rates of unfavourable outcomes in patients with COVID-19 treated with a three-day early course of remdesivir and who are at high risk for severe COVID-19 progression. First of all, we reported a progression of COVID-19 in only 8% of enrolled patients. Comparing these data with the PINETREE trial is difficult as in the trial a much lower percentage of COVID-19 related hospitalization in active arm (0.7%) was shown [6]. Actually, in the registration trial, a notable difference in enrolled patients compared to the available real-life studies was shown. In fact, in the PINETREE trial obesity and diabetes were the most frequent risk factors for severe COVID-19 (55.2% and 61.6%, respectively). Moreover, only 30.2% of enrolled patients were older than 60 years and only 4% were immunocompromised. In our study the most frequent risk factor for progression was immunodeficiency (58% of patients). Similarly, in a study by Piccicacco et al., 77% of patients were immunocompromised [13]. In this study they compared efficacy of early treatment for SARS-CoV-2 infection with remdesivir (82 patients) or sotrovimab (88 patients) to a control group (90 patients) in a matched retrospective cohort. Controls were defined as patients who refused the treatment or were not available to attend the scheduled visit. The authors showed that patients in the remdesivir and sotrovimab cohorts were less likely to be hospitalized compared with controls (11%, 8% and 23%, respectively) [13]. When comparing the two treatment groups, there was no difference in the rate of hospitalization between sotrovimab and remdesivir (*p* = 0.5) [13]. Moreover, we showed a similar rate of ADRs when compared to the data reported in the PINETREE trial. In fact, we reported at least one ADR in 13.3% of patients with no differences between the combination or the remdesivir alone group; similarly, in the trial the authors showed at least one ADR in 14% of treated patients [6]. Finally, we found no differences in clinical outcomes between patients treated with remdesivir alone and those treated with the combination of remdesivir and mAbs. However, the limited sample size precludes us to draw a definitive conclusion regarding the potential additive or synergistic effect between remdesivir and mAbs.

We acknowledge that the major limitation of our study is the lack of a control group. This is due to the prospective observational nature of the study and ethical issues. Actually, the percentage of hospitalization and/or worsening of respiratory conditions in our study was similar to the results by Piccicacco et al., despite the combined administration of monoclonal antibodies in half of our patients (seronegative patients). It is indeed noteworthy that immunosuppression and humoral non-response are associated with fatal outcomes of breakthrough infections [14]. In the second published study on early treatment with remdesivir, the authors focused on another group of immune compromised patients (Solid Organ transplant patients (SOTs)) and compared early remdesivir administration with no treatment. They overcame ethical issues as they enrolled in the control group patients who refused antiviral treatment or patients with eGFR less than 30 mL/min (which represents a contraindication for remdesivir administration). They enrolled 24 patients (7 cases and 17 controls) and 88% of controls were affected by severe chronic kidney disease (eGFR < 30 mL/min). The authors claimed that remdesivir administration had a significant effect on reducing the hospitalization rate and progression of COVID-19 (aHR 0.05 *p* = 0.01). However, it should be cleared that the control group had an underlying baseline higher risk of bad outcomes due to impaired renal function [15].

Another potential limitation of our study is represented by the difficulty in the interpretation of data among patients who mostly received SARS-CoV-2 vaccination. However, it must be noted that most of the included patients had immunodeficiency (58.1%). Most of these patients had a haematological disorder (e.g., non-Hodgkin lymphoma, acute leukaemia) and it is well-known that patients with a haematological disease show an impaired response to SARS-CoV-2 vaccination [16] and an increased rate of severe and fatal outcomes related to COVID-19 [17]. Finally, in a real-life study from a Mexican tertiary care centre, authors enrolled 126 patients (52 treated with remdesivir and 72 not treated). Characteristics of patients in the two groups were similar except for the age (controls were significantly older than cases, *p* < 0.001). Similar to other studies, treated patients were significantly less likely to die or to be hospitalized during the next 28 days (9.3% vs. 43.1%, *p* < 0.001) [18]. Certainly, comparisons of COVID-19 related outcomes between patients who received and did not receive antivirals would better shed the light on the entity of advantages conferred by early treatment with these drugs, thus including remdesivir. However, in the current clinical scenario, comparing our results with previous studies or historical cohorts would lead to significant interpretation biases. In fact, COVID-19 has progressively turned from a widespread disease capable of severe clinical pictures in almost all patients, to a medical reason of concern for frail subjects (i.e., elderly, immunocompromised patients, and those with significant comorbidities) and for unvaccinated people. Remdesivir, and other antivirals with activity against SARS-CoV-2, have been approved for treatment of patients at risk for severe disease progression. In real-life practice, this aspect led to the establishment of two different groups of patients: (i) those with no risk factors for COVID-19 severe progression, who are not treated with antivirals or other drugs directly active against SARS-CoV-2 and who only receive symptomatic treatments, and (ii) patients at risk for severe COVID-19 eligible for antivirals treatment. Thus, a comparison between patients treated and not treated with remdesivir should be made (in the real-life) between these two groups of patients, which are incomparable for clinical characteristics and rates of unfavorable outcomes, such as hospitalization, ICU admission and death. In order to overcome these limitations, we extracted data for comparisons from a government Italian dataset (available online: https://covid19.infn.it, accessed on 16 January 2023). According to these data, hospitalization rates among all patients with COVID-19 in Italy from 10 January 2022 to 31 March 2022, were 18.1% and 27.9% in patients aged >60 and >70 years, respectively. Despite the rawness of this comparison, it may support the use of remdesivir, also considering the peculiarity of our study’s population.

In conclusion, in our study we showed a lower hospitalization rate (8%) than the Mexican real-life study (13%) in which 82.2% of immunocompromised patients were enrolled. This result should be validated comparing real-life efficacy of different early treatments approaches (monoclonal antibodies, oral antivirals or remdesivir). In our study we also showed no differences between patients treated with remdesivir alone or a combination of remdesivir and monoclonal antibodies. However, patients in the combination group were more likely to have negative anti SARS-CoV-2 serology (37% vs. 10%) and showed higher CRP values and lower lymphocyte count, compared with patients treated with remdesivir alone. Despite these conditions have been already associated with severe outcome of COVID-19, in our study we showed a low progression rate also in these very high-risk patients treated with the combination. Since no strong conclusions can be drawn due to the lack of a control group, further larger studies are needed in order to define the best early treatment strategy in this setting and to assess the efficacy of the combination of antivirals and monoclonal antibodies in patients at highest risk of development of severe COVID-19 (i.e., immunocompromised patients and/or humoral non-responders to SARS-CoV-2 vaccination).

## Figures and Tables

**Table 1 vaccines-11-00200-t001:** Clinical characteristics of patients enrolled and differences between the two treatment groups.

	All Patients (n = 62)	Remdesivir (n = 30)	Remdesivir + mAb (n = 32)	*p*-Value
Male sex (n, %)	33 (53.2)	16 (53.3)	17 (53.1)	0.987
Age (years; median, IQR)	66 (49–76)	71 (58–77)	61 (49–70)	0.077
Age > 65 years (n, %)	18 (29.0)	9 (30.0)	9 (28.1)	0.871
Type of initial hospital admission (n, %)				0.465
- Outpatients	46 (74.2)	21 (70.0)	25 (78.1)	
- Inpatients	16 (25.8)	9 (30.0)	7 (21.9)	
MASS Score	5 (3–6)	5 (3–6)	5 (3–6)	0.201
Comorbidities (n, %)				
- Obesity	7 (11.3)	1 (3.3)	6 (18.8)	0.062
- CKD	3 (4.8)	1 (3.3)	2 (6.3)	0.525
- Diabetes	5 (8.1)	5 (16.7)	0 (0.0)	0.022
- Immunodeficiency	36 (58.1)	18 (60.0)	18 (56.3)	0.765
- Cardiovascular disease	14 (22.6)	12 (40.0)	2 (6.3)	0.001
- Chronic liver disease	1 (1.6)	1 (3.3)	0 (0.0)	0.484
- Chronic pulmonary disease	8 (12.9)	5 (16.7)	3 (9.4)	0.317
- Neurodegenerative disorder	3 (4.8)	1 (3.3)	2 (6.3)	0.525
N ° of comorbidities (n, %)				
- 1	34 (54.8)	16 (53.3)	18 (56.3)	0.818
- 2	19 (30.6)	11 (36.7)	8 (25%)	0.319
- 3	2 (3.2)	2 (6.7)	0 (0.0)	0.230
- >2	21 (33.9)	13 (43.3)	8 (25.0)	0.127
Symptoms at admission (n, %)				
- Fever	27 (43.5)	12 (40.0)	15 (46.9)	0.585
- Cough	20 (32.3)	10 (33.3)	10 (31.3)	0.861
- Ageusia/Anosmia	11 (17.7)	7 (23.3)	4 (12.5)	0.264
- Pharyngodynia	9 (14.5)	2 (6.7)	7 (21.9)	0.089
- Asthenia	17 (27.4)	10 (33.3)	7 (21.9)	0.312
- Headache	8 (12.9)	4 (13.3)	4 (12.5)	0.609
- Myalgia	9 (14.5)	4 (13.3)	5 (15.6)	0.543
- Gastrointestinal symptoms	1 (1.6)	1 (3.3)	0 (0.0)	0.484
- Dyspnoea	0 (0.0)	0 (0.0)	0 (0.0)	n/a
SARS-CoV-2 vaccinated (n, %)	47 (75.8)	25 (83.3)	22 (68.8)	0.180
- Vaccinated with positive serology	30 (48.4)	16 (53.5)	14 (43.8)	0.569
- Vaccinated with negative serology	8 (1.3)	1 (3.3)	7 (21.9)	0.033
- Vaccinated with unknown serology	10 (1.6)	8 (26.7)	2 (3.2)	0.032
SARS-CoV-2 serology at admission (n, %)				0.049
- Positive	34 (54.8)	17 (56.7)	17 (58.6)	
- Negative	15 (24.2)	3 (10.0)	12 (37.5)	
- Not known	13 (21.0)	10 (33.3)	3 (9.4)	
Laboratory exams (median, IQR)				
- CRP (mg/dL)	1.69 (0.65–7.06)	0.84 (0.45–6.93)	2.07 (0.80–7.20)	0.184
- WBC (cell/µL)	5130 (3562–8295)	6010 (3620–8270)	4770 (3390–8370)	0.703
- Lymphocytes (cell/µL)	965 (630–1370)	1170 (880–1840)	810 (510–1190)	0.025
- LDH (U/L)	229 (192–309)	209 (169–254)	254 (205–313)	0.094
Hospitalization (n, %) *	4 (8)	2 (9) °	2 (8) #	0.626
Increase in oxygen supplementation (n, %)	2 (3.2)	0 (0.0)	2 (6.3)	0.262
ICU admission (n, %)	1 (1.6)	0 (0.0)	1 (3.1)	0.516
Death (n, %)	1 (1.6)	0 (0.0)	1 (3.1)	0.516
Time from positive swab and treatment (days; median, IQR)	2 (1–3)	2 (1–3)	2 (2–3)	0.685
Time from symptoms onset and treatment (days; median, IQR)	3 (2–4)	3 (1–4)	4 (2–5)	0.183

* n = 46; ° n = 21; # n = 25.

**Table 2 vaccines-11-00200-t002:** Frequency of COVID-19 progression according to patients’ characteristics.

	Progression Yes (n = 5)	Progression No (n = 57)	*p*-Value
Male sex (n, %)	4 (80.0)	29 (50.9)	0.220
Age > 65 years (n, %)	1 (20.0)	17 (29.8)	0.545
≥2 comorbidities (n, %)	3 (60.0)	18 (31.6)	0.210
SARS-CoV-2 vaccinated (n, %)	4 (80.0)	43 (75.4)	0.651
Negative serology at admission (n, %)	2 (40.0)	13 (22.8)	0.349
CRP ≥ 6 mg/dL (n, %)	0 (0.0)	16 (30.8)	0.179
Lymphocyte ≤ 1000 cell/µL (n, %)	2 (40.0)	29 (54.7)	0.433
LDH ≥ 300 U/L (n, %)	1 (20.0)	14 (26.4)	0.614
Remdesivir monotherapy (n, %)	2 (40.0)	28 (49.1)	0.531

**Table 3 vaccines-11-00200-t003:** ADRs recorded in the study population and differences between the two treatment groups.

	All Patients (n = 62)	Remdesivir (n = 30)	Remdesivir + mAb (n = 32)	*p*-Value
At least one	9 (14.6)	4 (13.3)	5 (15.6)	0.543
Headache	4 (6.5)	0 (0.0)	4 (12.5)	0.064
Fever	4 (6.5)	3 (10.0)	1 (3.1)	0.282
Gastrointestinal disorders	0 (0.0)	0 (0.0)	0 (0.0)	-
Bradycardia	1 (1.6)	1 (3.3)	0 (0.0)	0.484
Other ADRs	0 (0.0)	0 (0.0)	0 (0.0)	-

## Data Availability

The data presented in this study are available on request from the corresponding author.

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
