# Peer review of "Remdesivir Alone or in Combination with Monoclonal Antibodies as an Early Treatment to Prevent Severe COVID-19 in Patients with Mild/Moderate Disease at High Risk of Progression: A Single Centre, Real-Life Study"

_vaccines, 2023, doi:10.3390/vaccines11020200_

Round 1
Reviewer 1 Report
This study data belongs to patients treated between 10th of January 2022 to 31st of March 2022 their dates, with 3-days course of 62 remdesivir. After this date range, there are many studies have been published on Remdesivir resistance in the SARS COV-2 VOC strains. I believe that the publication of the results of the study is delayed. In addition, I think that it would be risky to bring up the recommendations of the researchers for the treatment of SARS CoV-2 by using Remdesivir, since more studies are needed.
It is known that SARS-CoV-2 nsp12-RdRp S759A and V792I mutation homologs confer RDV resistance. These mutations are known to exist in the B.1.529 (Omicron) and their subvariant BA.1.1, which is the most common variant all over the world.
Some studies on this subject are listed below:
- De novo emergence of a remdesivir resistance mutation during treatment of persistent SARS-CoV-2 infection in an immunocompromised patient: a case report.
- Remdesivir resistance in transplant recipients with persistent COVID-19 DOI: 10.1093/cid/ciac769
- Mutations in the SARS-CoV-2 RNA-dependent RNA polymerase confer resistance to remdesivir by distinct mechanisms
- In Vitro Selection of Remdesivir-Resistant SARS-CoV-2 Demonstrates High Barrier to Resistance
- Exploring remdesivir resistance in COVID-19-infected transplant recipients.
Although Remdesivir was used in America at the beginning of the pandemic, its use has now been abandoned. This is a natural consequence of the studies listed above.
Therefore, the probability of the results of the study being put into effect was found to be very low. Additionally, researchers believe that the results of these studies are not sufficient and more studies should be done. Moreover, they have expressed the following in a negative sense regarding their results. “Since no strong conclusions can be drawn due to the lack of a control group further larger studies are needed in order to define the best early treatment strategy in this setting and to assess the efficacy of the combination of antivirals and monoclonal antibodies in patients at highest risk of development of severe COVID-19.” In short, they have been contradicted their own data.
Reviewer 2 Report
This study represents an attempt to demonstrate the disease-treatment value of remdesivir, the nucleotide prog-drug targeting the SARS-CoV-2 RNA polymerase. The prospective approach of the study is acknowledged and data collection per subject is appropriate and accurately presented. However, as the authors admit, the study is strikingly lacking any attempt at including an untreated control group/data, which is unfortunate in light of authors' highly meticulous recruitment, treatment protocols, and data collection. Their attempt to remedy this omission of control data by referencing at least 2 other studies falls short of supporting their claim the remdesivir appears to have some benefit. A suggestion to salvage this study for publication would be 1) to put in a tabular form the data of untreated (and treated) control groups from other published studies in comparison to that of this study, and 2) obtain and present historical data of untreated COVID subjects at the Infectious Diseases Unit of the University of Naples Federico II clinic where the current "treatment study" study was conducted.
Reviewer 3 Report
1. What is the novelty of this study?
2. What are the controls in this study?
3. What is the timeline for this study? Please include the details in the abstract.
4. There are no females included in this study, so the data represented have some bias. Please justify and discuss.
5. Did the authors collect information about the vaccination of the patients? Are they all vaccinated or not vaccinated?
6. What are the limitations of this study? Please include a separate section describing limitations.
7. Effects of Remdesivir and monoclonal antibodies were additive or synergistic? Please check this and provide the details.
8. What is the possible combined mechanism of action of both redeliver and monoclonal antibodies against COVID-19? Please discuss.
Round 2
Reviewer 1 Report
We know that in accordance with the vaccination campaign against Covid-19 in Europe and especially in Italy, a large proportion of the population is vaccinated. The hypothesis that the single or combined use of Remdesivir and/or MoAbs in treatment against this infection will reduce the rate of patients going to the intensive care unit is also highly unlikely to be evaluated independently of the vaccine efficacy.
Although the authors have answered some of my concerns regarding this issue, I still have my doubts due to the lack of control groups to determine whether this comparison is meaningful or not.
In this context, I recommend that the use of remdesivir alone in the treatment of the hypothesis of this study should not be highlighted. Moreover, antiviral use against covid 19 is no longer preferred in many countries.
Author Response
Point 1: We know that in accordance with the vaccination campaign against Covid-19 in Europe and especially in Italy, a large proportion of the population is vaccinated. The hypothesis that the single or combined use of Remdesivir and/or MoAbs in treatment against this infection will reduce the rate of patients going to the intensive care unit is also highly unlikely to be evaluated independently of the vaccine efficacy.
Although the authors have answered some of my concerns regarding this issue, I still have my doubts due to the lack of control groups to determine whether this comparison is meaningful or not.
In this context, I recommend that the use of remdesivir alone in the treatment of the hypothesis of this study should not be highlighted. Moreover, antiviral use against covid 19 is no longer preferred in many countries.
Response 1: We took the referee’s point, but, respectfully, we underline the following points:
- We believe that it is now impossible to design real-life studies without the bias of the high vaccination rate. In fact, in developed countries the rates of vaccination against SARS-CoV-2 are luckily very high. However, we claim that our real-life study can still add new information, since most of our patients had immunodeficiency. In detail, most of our patients had a haematological disorder (e.g., non-hodgkin lymphoma, acute leukaemia) and it is well-known that patients with a haematological disease show an impaired response to SARS-CoV-2 vaccination and an increased rate of severe and fatal outcome related to COVID-19. Moreover, we included a comparison with Italian government data on hospitalization rates among patients with COVID-19 in the same time period of our study, and we included the following sentence in the discussion: “In order to overcome these limitations, we extracted data for comparisons from a government Italian dataset (available at: https://covid19.infn.it). According to these data, hospitalization rates among all patients with COVID-19 in Italy from 10th of January 2022 to 31st of March 2022, were 18.1% and 27.9% in patients aged >60 and >70 years, respectively. Despite the rawness of this comparison, it may support the use of remdesivir, also considering the peculiarity of our study’s population.”
- Our real-life study was not designed to compare the efficacy between patients treated with remdesivir alone versus those treated in combination with monoclonal antibodies. Instead, the aim of our study was to describe the rates of unfavorable outcome in a real-life cohort of patients with severe comorbidities treated with remdesivir, regardless of the use of a monoclonal antibody in a combination treatment.
- We do not understand the referee’s sentence “antiviral use against covid 19 is no longer preferred in many countries”. To our knowledge, antivirals still represent the main treatment in the first days after the diagnosis of COVID-19 in patients at risk for severe disease progression (i.e. patient with with immunodepression, severe chronic diseases, obesity). regardless of vaccination status. We have already discussed this point in our previous rebuttal letter. Moreover, a recent study published on the New England Journal of Medicine showed that Remdesivir retain activity also in SARS-CoV-2 variants of recent occurrence (https://www.nejm.org/doi/full/10.1056/NEJMc2214302)
We included the following sentence in the discussion section:
- Another potential limitation of our study is represented by the difficulty in the interpretation of data among patients who mostly received SARS-CoV-2 vaccination. However, it must be noted that most of the included patients had immunodeficiency (58.1%). Most of these patients had a haematological disorder (e.g., non-Hodgkin lymphoma, acute leukaemia) and it is well-known that patients with a haematological disease show an impaired response to SARS-CoV-2 vaccination [16] and an increased rate of severe and fatal outcome related to COVID-19 [17]
Reviewer 2 Report
While the authors have attempted to clarify their position on the scope/objective of the manuscript with respect to the lack of a control reference group, the major problem still remains; i.e., there is no way to demonstrate that remdesivir had any clinical value. Merely claiming that this study is "descriptive" in nature does not warrant publication.
Author Response
Point 1: While the authors have attempted to clarify their position on the scope/objective of the manuscript with respect to the lack of a control reference group, the major problem still remains; i.e., there is no way to demonstrate that remdesivir had any clinical value. Merely claiming that this study is "descriptive" in nature does not warrant publication.
Response 2: We still believe that results from real-life cohort can really help clinicians in their daily clinical practice. We reported rates of unfavorable outcomes among patients with COVID-19 at high risk for severe disease progression. We claim that most of our patients had immunodeficiency. Actually, most of them had a hematological disease. We are confident about the novelty of our data, and we are honestly sure that no real-life studies without biases can be design in this particular population. Please note that we made improvement in our discussion. In particular, we included the following sentence:
“In order to overcome these limitations, we extracted data for comparisons from a government Italian dataset (available at: https://covid19.infn.it). According to these data, hospitalization rates among all patients with COVID-19 in Italy from 10th of January 2022 to 31st of March 2022, were 18.1% and 27.9% in patients aged >60 and >70 years, respectively. Despite the rawness of this comparison, it may support the use of remdesivir, also considering the peculiarity of our study’s population”
Reviewer 3 Report
The authors successfully responded and updated the manuscript as per the reviewer's suggestions.
Author Response
Thank you very much. Please note that we also improved the discussion section including the following sentence:
“In order to overcome these limitations, we extracted data for comparisons from a government Italian dataset (available at: https://covid19.infn.it). According to these data, hospitalization rates among all patients with COVID-19 in Italy from 10th of January 2022 to 31st of March 2022, were 18.1% and 27.9% in patients aged >60 and >70 years, respectively. Despite the rawness of this comparison, it may support the use of remdesivir, also considering the peculiarity of our study’s population”
Round 3
Reviewer 1 Report
The authors responded to all my criticisms.
Reviewer 2 Report
The authors have sufficiently clarified the "safety objective" of the study and improved the Discussion by addressing the possible efficacy of Remdesivir by benchmarking it against published control rates. As the authors stated, the value of safety outcomes to other clinicians using the drug is of importance and would warrant publication.